# Risk Factors Contributing to Reinfection by SARS-CoV-2: A Systematic Review

Walter Gómez-Gonzales [1,*], Luis A. Chihuantito-Abal [2], Carlos Gamarra-Bustillos [3], Julia Morón-Valenzuela [1], Jenny Zavaleta-Oliver [4], Maria Gomez-Livias [3], Lidia Vargas-Pancorbo [5], María E. Auqui-Canchari [6] and Henry Mejía-Zambrano [4]

[1] Escuela de Medicina, Filial Ica, Universidad Privada San Juan Bautista, Ica 11001, Peru; julia.moron@upsjb.edu.pe

[2] Facultad de Ciencias de la Salud, Universidad Andina del Cusco, Cuzco 08006, Peru; lchihuantito@uandina.edu.pe

[3] Escuela de Medicina, Universidad Norbert Wiener, Lima 15046, Peru; carlos.bustillos@uwiener.edu.pe (C.G.-B.); a2020104983@uwiener.edu.pe (M.G.-L.)

[4] Escuela de Medicina Humana, Universidad Privada San Juan Bautista, Lima 15067, Peru; jenny.zavaleta@upsjb.edu.pe (J.Z.-O.); henry.mejia@upsjb.edu.pe (H.M.-Z.)

[5] Escuela Posgrado, Universidad San Antonio Abad del Cusco, Cusco 08006, Peru; lidia.vargas@unsaac.edu.pe

[6] Facultad Ciencias de la Salud, Universidad Tecnologica del Peru, Lima 15046, Peru; maria.auqui@utp.edu.pe

* Correspondence: walter.gomez@upsjb.edu.pe

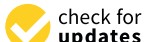



**Highlights:**

**What are the main findings?**

- This comprehensive analysis meticulously examined 51 studies, pinpointing 27 that rigorously adhered to stringent criteria. The incorporation of diverse studies afforded a panoramic perspective on COVID-19 reinfection.
- Emphasis was placed on the effectiveness of vaccination, showcasing a remarkable risk reduction of up to 66% with the administration of two vaccine doses.

**What is the implication of the main finding?**

- In spite of an overall low reinfection rate, the findings unveiled noteworthy patterns. Factors such as non-vaccination, advanced age, and the presence of comorbidities emerged as significant contributors to the likelihood of reinfection.
- While the review furnished valuable insights, it underscored the persistent necessity for more nuanced observational studies. Special attention was urged, particularly regarding emerging variants and the duration of immunity post-infection or post-vaccination.

**Abstract:** This article aims to systematize the evidence regarding risk factors associated with COVID-19 reinfection. We conducted a systematic review of all the scientific publications available until August 2022. To ensure the inclusion of the most recent and relevant information, we searched the PubMed and Scopus databases. Thirty studies were reviewed, with a significant proportion being analytical observational case-control and cohort studies. Upon qualitative analysis of the available evidence, it appears that the probability of reinfection is higher for individuals who are not fully immunized when exposed to a new variant, females, those with pre-existing chronic diseases, individuals aged over 60, and those who have previously experienced severe symptoms of the disease or are immunocompromised. In conclusion, further analytical observational case-control studies are necessary to gain a better understanding of the risk factors associated with SARS-CoV-2 (COVID-19) reinfection.

**Keywords:** risk factors; reinfection; COVID-19; SARS-CoV-2; public health

## 1. Background

SARS-CoV-2, the causative agent of COVID-19, has precipitated a global pandemic [1]. The immune system responds swiftly in infectious processes, and the infection by SARS-CoV-2 (COVID-19) is no exception. Studies from early 2021 initially suggested natural immunity could protect against reinfection for at least 8 to 12 months [2,3]. However, real-world cases have contradicted this, reporting reinfection occurring within 3 to 6 months [4,5]. Furthermore, the understanding of sustainable, long-lasting protective immunity post-COVID-19 infection remains uncertain, and the underlying mechanisms are not yet fully comprehended [6,7]. Immunity following infection is often established through immune responses involving IgG antibodies and orchestrated by specialized T cells. When assessing post-infection immunity, primary considerations encompass the identification of protective functions, delineation of measurable biological markers, and precise definition of terms, such as reinfection, recurrence, readmission, mortality, and transmission to others [8].

In the initial phase of the 2020 pandemic, the first documented instances of SARS-CoV-2 reinfection emerged in Hong Kong. The interval between the two episodes was observed to be 142 days, marked by mild symptoms during the initial infection and an absence of symptoms in the subsequent episode [9]. This discovery prompted the identification of suspected and confirmed cases of reinfection globally [10].

The current concept of reinfection lacks consistency [11]. The European Center for Disease Prevention and Control defines reinfection as confirmed by laboratory studies of two infections by different strains with a minimum distance, as supported by phylogenetic and epidemiological evidence [12]. Vaccination against SARS-CoV-2 has shown promise in reducing infection rates, but there is still ambiguity regarding reinfection cases among fully vaccinated, partially vaccinated, and unvaccinated individuals. The primary obstacle in probing leaky protection within SARS-CoV-2 immunity lies in the inherent difficulty in measuring the viral dose, whether incident or cumulative over time. Current investigations often resort to assessing proxies, such as the proximity and duration of exposure to an infected index case. However, the utility of these proxies is constrained by the scarcity of reliable information at the necessary scales and the potential for misclassification owing to movement and social interactions in real-world settings [13].

Despite explicit safety measures recommending full vaccination, mask-wearing, and social distancing for all, including rehabilitated patients, there are studies suggesting the possibility of reinfection among this group [14]. An intriguing study conducted within a prison population underscores the imperative for stratified interventions to curtail the spread of SARS-CoV-2, particularly in dense settings, such as congregate environments. This necessity is emphasized in situations where prolonged contact is probable, such as households with infected individuals [13].

Reports of possible reinfections by COVID-19 after the initial recovery have increased over time [15]. Recent studies have highlighted the unknown degree of protective immunity conferred by SARS-CoV-2 infection, making the understanding of reinfection possibilities crucial [16]. This comprehension is pivotal for informing global health and government policies and may contribute to evaluating the feasibility of the concept of herd immunity, a topic discussed by some scientists [17–19].

Early risk stratification in COVID-19 remains challenging [7]. Thus, understanding reinfection by COVID-19 becomes a critical element guiding government and public health policies [6,20]. The present study aims to systematically present the most relevant evidence on the risk factors associated with COVID-19 reinfection. Investigating factors linked to COVID-19 reinfections is essential for assessing the past pandemic's landscape, as ignorance on this matter could complicate vaccine development and perpetuate viral outbreaks worldwide. This study addresses the existing gaps in the understanding of COVID-19 reinfections by conducting a systematic review of currently available clinical cases.

## 2. Methodology

This systematic review adhered to the guidelines outlined in the preferred reporting items for systematic reviews and meta-analyses (PRISMA) protocols [21].

### 2.1. Bibliographic Search Strategy

The selection of scientific articles involved a comprehensive search of research articles within the databases of the PubMed and Scopus repositories. To structure the bibliographic search strategy, specific descriptors related to factors such as reinfection, COVID-19, and SARS-CoV-2 were employed. The search was confined to studies published from 2019 onward, coinciding with the occurrence of the first COVID-19 case. Inclusivity extended to all the studies providing substantial data on COVID-19 reinfection, without imposing restrictions based on language, age, or sex (Table S1). In this investigation, COVID-19 reinfection was delineated as the emergence of a new infection after the declaration of recovery from a prior infection. References within the included articles were scrutinized as potential sources for additional studies.

### 2.2. Eligibility Criteria

The nature of the study guided a systematic review following specific criteria. The inclusion criteria encompassed analytical observational designs, including cases and controls, cohorts, and, when necessary, descriptive studies. It is important to highlight that case reports were deliberately excluded from the systematic review. Furthermore, the study emphasized the inclusion of evaluation studies that investigated elements associated with risk factors for COVID-19 reinfection or studies focused on the recurrence of the disease.

### 2.3. Selection of Studies and Data Extraction

The titles and abstracts of the pertinent articles, extracted from the databases, were transferred to the Rayyan online software for the systematic organization of the literature search results. Following the elimination of duplicates, the process for selecting articles based on the title and abstract was conducted independently by two reviewers (WGG and HMZ). Any disagreements were resolved through discussions to achieve consensus. Subsequently, the ultimate inclusion of articles was determined based on the full texts, independently assessed by all the authors.

### 2.4. Summary of Results

The outcome of this review enabled a qualitative analysis of factors associated with COVID-19 reinfection.

### 2.5. Synthesis of Results

A formal narrative synthesis of the gathered data was performed, while a formal statistical synthesis was not pursued. Syntheses primarily focused on qualitatively analyzing clinical manifestations mentioned in each country of the published studies.

### 2.6. Study Quality Assessment

Given the enduring nature of the COVID-19 pandemic, our focus was on studies conducted between 2020 and 2022, a period marked by the highest concentration of cases. The quality assessment of the included articles was independently conducted by two reviewers, CGB and MGL, utilizing established criteria. By employing the Joanna Briggs Institute critical appraisal tools for use in JBI systematic reviews, studies were evaluated for methodological quality and assigned a score of either present (1) or absent (0), which was then tallied to derive a final value [22]. The two reviewers reached a consensus on the quality assessment results through discussion. The Newcastle-Ottawa scale (NOS) was used to analyze the risk of bias [23].

### 3. Results

In the initial phase of the literature search, a total of 2777 records were identified. After the removal of 2523 duplicate entries, a subsequent screening process, which involved assessing titles and abstracts, resulted in 254 articles under consideration. Following this, 203 articles were excluded, leaving 51 studies for an in-depth evaluation of their full texts. Following a meticulous review, 30 articles were ultimately determined to meet the pre-established inclusion criteria and were subjected to detailed analysis as a part of the systematic review (Figure 1).

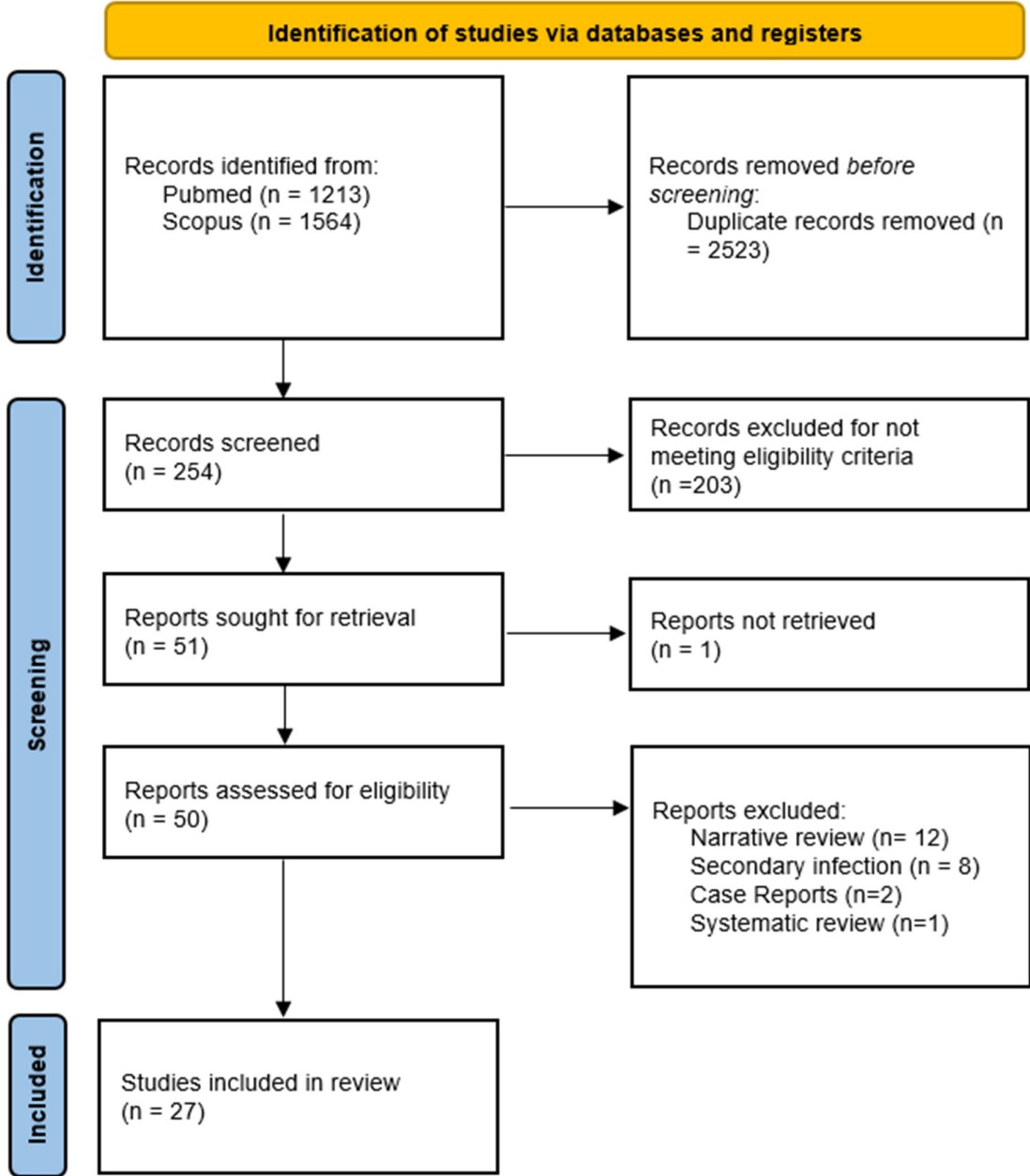

**Figure 1.** PRISMA flowchart to show the study selection process.

The systematization of the analyzed articles is shown in Table 1.

**Table 1.** Analytical observational studies from the beginning of COVID-19 to August 2022.

| No. | Country | Publication Date | Author | Title | Findings |
|---|---|---|---|---|---|
| 1 | Italy | May 2022 | Sacco et al. [24] | Risk and protection factors for SARS-CoV-2 reinfections | Not being vaccinated emerges as the most significant risk factor. The risk of infection escalates 18 times more with the Omicron variant. Individuals who have experienced a severe initial infection and are over 60 years of age face heightened risks. |
| 2 | Italy | May 2022 | Flacco et al. [25] | Risk of SARS-CoV-2 reinfection 18 months after primary infection: a population-based observational study | Risk is significantly higher in women, young people, and people who have not been vaccinated. |
| 3 | Saudi Arabia | April 2022 | Shaheen et al. [26] | COVID-19 reinfection: a multicenter retrospective study in Saudi Arabia | Among the 35,288 analyzed patients, 0.37% experienced reinfection. The mean age of the participants was $40.95 \pm 19.48$ years, and 50.76% were female. |
| 4 | Sweden | April 2022 | Nordström et al. [27] | Risk of SARS-CoV-2 reinfection and hospitalization from COVID-19 in people with natural and hybrid immunity: a retrospective total population cohort study in Sweden | Two doses of the SARS-CoV-2 vaccine were associated with a 66% lower risk of reinfection. |
| 5 | USA | August 2021 | Slezak et al. [28] | Rate and severity of suspected reinfection by SARS-CoV-2 in a cohort of PCR-positive COVID-19 patients | Out of 75,149 cases, only 315 suspected reinfections were identified. Significant independent predictors of suspected reinfection included being a woman, an adult, being immunocompromised, and having previously been hospitalized for COVID-19. |
| 6 | England | April 2022 | Mensah et al. [29] | Illness severity during SARS-CoV-2 reinfection: a nationwide study | The female sex represented the highest proportion of reinfections (67%). Evidence suggested a rising increase in the risk of infection in individuals over 70 years of age. |
| 7 | China | December 2020 | Yao et al. [30] | Factors associated with a recurrence of SARS-CoV-2 after hospital discharge among patients with COVID-19: systematic review and meta-analysis | The primary factors associated with the recurrence of COVID-19 after hospital discharge included advanced age, severity of the previous infection, bilateral pulmonary infiltration, and decreases in leukocyte, platelet, and CD4+ T counts. |
| 8 | South Africa | March 2022 | Pulliam et al. [31] | Increased risk of reinfection by SARS-CoV-2 associated with the emergence of Omicron in South Africa | The relative risk of reinfection by COVID-19 increased with the third wave, marked by the emergence of the Omicron variant. |
| 9 | Bahrain (Middle East) | August 2022 | Almadhi et al. [3] | Epidemiological evaluation of reinfection by SARS-CoV-2 | The proportion of reinfected males was significantly higher at 60.3% ($p < 0.0001$), particularly within the 30–39 age group (29.7%). The lowest number of episodes occurred between 3 and 6 months after the first infection (20.6%), while the highest number of episodes occurred from the 9th month after the previous infection (46.4%). |
| 10 | Saudi Arabia | July 2022 | Al-Otaiby et al. [32] | SARS-CoV-2 reinfection rate and outcomes in Saudi Arabia: a national retrospective study | In the analysis of the risk factors, reinfection was highly associated with comorbidities, including HIV, obesity, and being healthcare personnel. |
| 11 | Mexico | April 2021 | Murillo-Zamora et al. [33] | Predictors of severe laboratory-confirmed symptomatic SARS-CoV-2 reinfection | Factors associated with an increased risk of severe symptomatic SARS-CoV-2 reinfection included a history of laboratory-confirmed severe coronavirus disease. |
| 12 | Mexico | May 2021 | Garduño-Orbe et al. [34] | SARS-CoV-2 reinfection in health workers in Mexico: Case report and review of the literature | Two out of the four cases of reinfection were severe, while in the remaining cases, the clinical manifestations resembled those of the previous infection. |

**Table 1.** *Cont.*

| No. | Country | Publication Date | Author | Title | Findings |
|---|---|---|---|---|---|
| 13 | Spain | January 2022 | Sánchez-Varela et al. [35] | Reinfection by the Omicron variant in patients previously performed with the Delta variant of the SARS-CoV-2 coronavirus: an increasingly frequent reality in primary care | Increased incidence due to the Omicron variant |
| 14 | China | May 2020 | Hu et al. [36] | Recurrent positive reverse transcriptase–polymerase chain reaction results for coronavirus disease 2019 in patients discharged from a hospital in China | There were no significant differences between the demographic and baseline clinical characteristics in the recurrence and non-recurrence groups. |
| 15 | Republic of Cyprus | March 2022 | Quattrocchi et al. [37] | Effect of vaccination on the risk of reinfection by SARS-CoV-2: a case-control study in the Republic of Cyprus | Their findings support the benefit of vaccination for individuals previously infected with SARS-CoV-2. |
| 16 | USA | August 2021 | Cavanaugh et al. [38] | Reduced risk of reinfection with SARS-CoV-2 after vaccination against COVID-19 in Kentucky, May–June 2021 | Those not vaccinated were from 2.3 to 4 times more likely to be reinfected compared to those who received all vaccinations (odds ratio (OR) = 2.34; 95% confidence interval (CI) = 1.58–3.47). In individuals with prior infection, full vaccination provided additional protection against reinfection. |
| 17 | Serbia | July 2022 | Medić et al. [39] | Risk and severity of SARS-CoV-2 reinfections during 2020–2022 in Vojvodina, Serbia: a population-level observational study | Those who were not vaccinated (OR = 1.23; 95% CI = 1.14–1.33), those with incomplete vaccination (OR = 1.33; 95% CI = 1.08–1.64), or fully vaccinated individuals (OR = 1.50; 95% CI = 1.37–1.63) were more likely to experience reinfection compared to patients who received the booster dose. |
| 18 | Austria | February 2021 | Pilz et al. [40] | Risk of SARS-CoV-2 reinfection in Austria | A relatively low risk of reinfection was documented. Patients with reinfections were of both sexes, spanning a wide age range, and were hospitalized during both infections. |
| 19 | Brazil | February 2021 | Adrielle Dos Santos et al. [41] | Recurrent COVID-19 including evidence of reinfection and increased severity in thirty Brazilian healthcare workers | Out of 33 patients with recurrent COVID-19, 26 were women, and 30 were healthcare workers. The median time to recurrence was 50.5 days. |
| 20 | Sweden | January 2020 | Havervall et al. [42] | Robust humoral and cellular immune responses and low risk of reinfection at least 8 months after asymptomatic or mild COVID-19 | The presence of IgG anti-spike antibodies is associated with a significantly reduced risk of reinfection for up to 9 months following asymptomatic or mild COVID-19. |
| 21 | India | January 2022 | Nisha et al. [43] | Incidence of SARS-CoV-2 infection, reinfection, and post-vaccination and associated risks in healthcare workers in Tamil Nadu: a retrospective cohort study | Men and primary care providers were at a higher risk of infection. Partial vaccination status was identified as one of the determinants of reinfection. |
| 22 | USA | May 2022 | Levin-Rector et al. [44] | Reduced odds of SARS-CoV-2 reinfection after vaccination among New York City adults, July–November 2021 | Vaccination decreased the likelihood of reinfections, particularly when the Delta variant predominated. |
| 23 | Several countries | August 2021 | Sotoodeh Ghorbani et al. [8] | Epidemiological characteristics of cases with reinfection, recurrence, and hospital readmission due to COVID-19: a systematic review and meta-analysis | The recurrence of reinfections was higher in women in comparison to men. Hospital readmission rates were similar for both sexes. There remains uncertainty regarding long-term immunity after SARS-CoV-2 infection. |
| 24 | France | March 2022 | Nguyen et al. [45] | SARS-CoV-2 reinfection and severity of COVID-19 | Reinfection represented 0.4% of the diagnosed positive cases. Among the 64 patients who underwent serological tests, 39 had antibodies when sampled early in their second infection. Only seven patients (3.4%) experienced infection twice with the same variant. |

**Table 1.** *Cont.*

| No. | Country | Publication Date | Author | Title | Findings |
|-----|---------|------------------|--------|-------|----------|
| 25 | England | March 2022 | Mensah et al. [46] | Risk of SARS-CoV-2 reinfections in children: a prospective national surveillance study between January 2020 and July 2021 in England | Children were at a lower risk of reinfection compared to adults, and reinfections were not associated with more severe disease or fatal outcomes. |
| 26 | Mexico | June 2022 | Ochoa-Hein et al. [47] | Significant increase in SARS-CoV-2 reinfection rate in vaccinated hospital workers during the Omicron wave: a prospective cohort study | The SARS-CoV-2 reinfection rate increased significantly during the Omicron wave despite a high primary vaccination coverage rate. Nearly a third of reinfected individuals had received a booster vaccine at least 14 days before the last episode of COVID-19. |
| 27 | South Korea | August 2022 | Jang et al. [48] | SARS-CoV-2 reinfection in general population, South Korea: nationwide retrospective cohort study | Individuals with one dose of the vaccine had the highest reinfection rate at 642.2 per 100,000, followed by those not vaccinated (536.2/100,000) and individuals vaccinated with two doses (406.3/100,000). |

Source: Self-prepared based on the systematic review.

A total of 27 articles were included in this systematic review, with 20 corresponding to case-control studies and cohorts. Most of the reviewed literature originated from European countries, indicating a biomedical research focus on the relationship between previous COVID-19 episodes and reinfection by SARS-CoV-2. Following Europe, studies from America, Asia, the Middle East, and other locations were considered.

The evaluation of the manuscript quality was conducted using the Joanna Briggs Institute critical appraisal tools designed for JBI systematic reviews, and the comprehensive results are outlined in Table S2. Additionally, an in-depth analysis of the risk of bias using the NOS scale is meticulously presented in Table S3.

A commonality among the selected studies is the association with intrinsic aspects, such as non-immunization, incomplete immunization, and their correlation with the female gender. In the analyzed cases, a prevailing element is the low rates of reinfection.

Concerning associated risks in healthcare workers during the COVID-19 pandemic, it was determined that intrahospital infections and reinfections were mostly associated with the Omicron variant, with few cases linked to the Delta variant [42,44] in different geographical locations.

Factors contributing to greater recurrence in the analyzed studies include aspects related to immunocompromised patients and an increase in age. The risk, with or without booster doses, tends to rise with age. Comorbidities, such as obesity, asthma, HIV, and diabetes mellitus, were also significant, with the association of more than two factors increasing the risk of reinfection.

For the unvaccinated, studies revealed reinfection probabilities ranging from 2 to 4 times, establishing non-vaccination as a significant risk factor. This risk was consistently elevated in the female gender across various locations, indicating a gender-based propensity for reinfection. The reviewed studies concurred in stating that individuals with only one vaccine dose exhibited the highest reinfection rates. In contrast, those with two vaccine doses had an average of a 66% lower risk of reinfection. It was also affirmed that the female gender manifested a higher recurrence rate. Overall, vaccination demonstrated a positive and significant effect against potential reinfection.

Additionally, it was disclosed that children had a lower probability of reinfection compared to adults. These findings collectively contribute to our understanding of the complex dynamics of COVID-19 reinfections and highlight the protective impact of vaccination across different demographics.

## 4. Discussion

Based on the findings from the analyzed studies, definitive conclusions regarding an increased probability of reinfection in relation to factors such as incomplete immunization and exposure to new variants are challenging to draw [35]. However, being of the female gender, having a pre-existing chronic disease, belonging to the age group of over 60 years, and having previously experienced severe symptoms associated with COVID-19, or in some cases, being immunocompromised, can be considered as factors associated with reinfection [8,11,20,30]. Moreover, the simultaneous presence of more than one of these factors appears to exacerbate the risk of reinfection. It is crucial to note that there is substantial evidence, as supported by the reviewed medical literature, suggesting a high likelihood of association between the aforementioned factors and cases of reinfection by SARS-CoV-2. Additionally, cases linked to the Delta and Omicron variants were directly related to the analyzed factors.

Although China plays a significant role in the pandemic, some studies from there were excluded owing to the language barrier because they were published only in Mandarin Chinese or the equivalent. China's experience, characterized by stringent control measures and a proactive research approach, holds valuable lessons [30,36]. The swift execution of containment protocols and thorough surveillance has the potential to unveil successful strategies applicable on a global scale. The consideration of these studies in future research is recommended for a more comprehensive and objective understanding of reinfection cases by COVID-19.

Delving deeply into the mechanisms that underpin the efficacy of vaccines in preventing reinfections emerges as a pivotal stride in grasping the intricacies of immunization dynamics [37,38]. Substantial variations in effectiveness have become evident across diverse vaccine types and dosing regimens. mRNA-based vaccines, exemplified by those developed by Pfizer-BioNTech and Moderna, have shown noteworthy efficacy, particularly in countering the challenges posed by emerging variants. In contrast, viral vector vaccines, such as those produced by AstraZeneca and Johnson & Johnson, have yielded somewhat distinct outcomes [13,43,44,47]. The issue of waning effectiveness over time has garnered significant attention. Several studies propose a gradual attenuation of protective measures over time, sparking dialogs on the necessity of booster doses. These insights underscore the critical importance of continuous monitoring and the adaptive refinement of vaccination strategies in response to the ever-evolving epidemiological landscape and the nuanced dynamics of immune responses as time unfolds.

Based on the reviewed literature, it can be concluded that to date, instances of potential reinfection by COVID-19 have been observed, albeit at a low overall rate of reinfection across various global locations. Access to updated data has the potential to offer a more precise understanding of regional trends, enabling a more thorough evaluation of how factors such as socioeconomics and public health strategies interact with the incidence of reinfections [3,25,32]. This detailed exploration goes beyond enhancing the accuracy of our assessments—it becomes a pivotal step toward tailoring control measures more effectively. Moreover, it unveils insights into local determinants that may play a crucial role in shaping the incidence of reinfections within the ongoing context of the pandemic. In essence, staying abreast of the latest data not only refines our strategic responses but also unveils nuanced perspectives that are invaluable in navigating the complexities of the current public health landscape [12,28,33].

Despite the valuable insights provided by the studies included in this systematic review, it is important to acknowledge and address their inherent limitations. One notable concern is the potential for bias in study selection, given that most of the studies originate from European countries, raising the possibility of geographical bias in the results. Additionally, the diverse definitions of reinfection across these studies may have affected the data homogeneity, making direct comparisons challenging. Variations in inclusion and exclusion criteria, as well as methodological differences, could also introduce systematic biases. These potential limitations underscore the need for cautious interpretation of the

results and emphasize the significance of future research to reconcile methodological variations, ensuring a more nuanced and comprehensive understanding of COVID-19 and SARS-CoV-2 reinfection rates.

To enhance our understanding of the risk factors associated with SARS-CoV-2 reinfection, there is a clear need for more studies employing an observational analytical design, including cases and controls. Molecular studies on the viral genome and its variants, as well as investigations into immune responses stratified by age, gender, and geographic location should not be overlooked. These complementary analyses can provide more precise data, contributing to a comprehensive understanding of the current landscape surrounding cases of COVID-19 and reinfection by SARS-CoV-2.

## 5. Conclusions

In conclusion, this systematic exploration is designed to unravel the intricate mechanisms associated with COVID-19 and SARS-CoV-2 on a global scale. Anticipating future research endeavors, we expect ongoing efforts to enhance and refine our understanding, offering progressively more accurate data in this dynamic and evolving field.

**Supplementary Materials:** The following supporting information can be downloaded at: https://www.mdpi.com/article/10.3390/arm91060041/s1, Table S1: Search strategy; Table S2: JBI assessment results of studies; Table S3: Newcastle–Ottawa Scale (NOS) Risk of Bias Assessment Results.

**Author Contributions:** The conception and design of the study were orchestrated by W.G.-G. and H.M.-Z., with meticulous data collection executed by L.A.C.-A., C.G.-B., J.M.-V. and J.Z.-O. The nuanced analysis and interpretation of the data were spearheaded by W.G.-G. and H.M.-Z. The manuscript preparation was skillfully undertaken by M.G.-L. and L.V.-P., while M.E.A.-C. and W.G.-G. provided critical intellectual contributions during the manuscript's review. All authors have read and agreed to the published version of the manuscript.

**Funding:** This research received no external funding.

**Institutional Review Board Statement:** Given that this study entailed a systematic review of pre-existing data without direct engagement with human participants or animals, ethical approval was deemed unnecessary. Despite the absence of formal ethical clearance, we emphasize that this systematic review adhered rigorously to the highest ethical standards, upholding a steadfast commitment to ethical principles throughout its execution.

**Informed Consent Statement:** Not applicable.

**Data Availability Statement:** Upon a reasonable request, the corresponding author can provide the study's data.

**Conflicts of Interest:** The authors affiliated with this systematic review openly declare the absence of any conflict of interest, affirming the impartiality and integrity of the research process.

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
