# Peer review of "Risk Factors Contributing to Reinfection by SARS-CoV-2: A Systematic Review"

_arm, doi:10.3390/arm91060041_

Round 1
Reviewer 1 Report
Comments and Suggestions for Authors
Thank you for allowing me to review this work. Although we are no longer in a pandemic situation, the work on COVID-19 is still very relevant.
In my opinion, the paper has serious problems to be published as it currently stands. My suggestions for improvement are as follows
a) Introduction:
References do not follow journal standard (no superscripts, brackets).
Line 29. First time you use an acronym, you should define it.
Line 34: What does it mean that the time to reinfection has been reduced from 3 to 6 months?
Line 35: Where are references 2 and 3?
Line 43-44: The sentence "Although it has been shown that vaccination against SARS-CoV-2 tends to reduce the cases of suffering from the infection." It seems to be unfinished. The same is true in the sentence on lines 48-51 (semicolon left over?).
Line 52: SARS-COV-26?
Lines 55-56: The acronym has been used in the introduction several times before. You should define it the first time you use it and not here. Same on line 68.
Line 62: The cited reference does not talk about controversies about herd immunity, but racial/ethnic inequalities in herd immunity. If it mentions the controversy, it should cite articles in which it is discussed.
The last paragraph of the introduction needs deep analysis and rephrasing. There are repeated ideas and confusing wording. There is not currently a pandemic situation, although COVID-19 studies continue to be very relevant.
b) Methodology
I do not consider that the term "factors" alone is part of a good search strategy. If the system (PRISMA in this case) is to ensure the reproducibility of the review, it should accompany the equations as you have used them, with the Boolean operators you have used. I suggest you add them in supplementary material.
If you have searched Pubmed (Medline) and Scopus, the third criterion is not necessary.
You should indicate in section 2.6 which evaluation scale or scales you have used to analyze the quality of the articles admitted for review.
c) Results
It is imperative that you begin the results section with the PRISMA 2020 article search flowchart, clearly indicating the reasons for rejection of the discarded full-text articles.
The quality of evidence is a further step in research that involves an evaluation applied after the articles have been selected based on the inclusion/exclusion criteria. Therefore, it is necessary that they choose and define in the methodology (section 2.6) which assessment scale or scales they have used and provide evidence of the assessment of the quality of the articles. This can also be included in supplementary material. This assessment should be transparent, and all dimensions of the assessment should be accessible to readers (see, for example, other systematic reviews, pages 114-132 of this source https://www.thelancet.com/cms/10.1016/S0140-6736(17)32802-7/attachment/4a1a9568-93f3-4b83-b8d6-6209b2808bdc/mmc1.pdf, or Table 4 of this source https://www.frontiersin.org/articles/10.3389/fpubh.2022.942595/full#T4).
Discussion
The data presented in Table 1 (references 13-42) are not integrated in the "Discussion" section; it is not clear why the authors went to such great lengths to review all these papers if the results are not adequately compared.
Comments on the Quality of English Language
The work needs to be proofread by a specialized translator.
Author Response
Thanks to the reviewer for his invaluable suggestions. The answer to each of your considerations is found in the attached file.

Reviewer 2 Report
Comments and Suggestions for Authors
Dear authors,
in my opinion the scientific soundness of your work is of great importance but the way you presented is not really clear. There are many sentences too long and with no punctuation with make the text difficult to understand.
I have some point to highlight:
line33-35: protection at 8 and 12 months I imagine are data derived from scientific works, as information of reinfection after 3 months. The authors statements: " not tell the truth" appears to strong in my opinion.
line 52: Typo in: CoV-26
line 71-74: IN my opinion this sentence need to be rephrased.
line 94-96: "in which data such as:....." Is missing a verb.
line 110: after "reinfection" there is a dot to be removed.
line 122: maybe is missing a "in" before China
line 125: A dot is missing after language and it should be: "will be taken into account"
line 132-133: "resulted in low rates of infections", for vaccinated or unvaccinated? with one dose or with booster? Is not clear.
line 149-153: This sentence is too long and without punctuation.
line 180: "precision" need a capital letter.
Comments on the Quality of English LanguageThe paper is not very well written, in same part it is difficult to understand. Same sentences are too long and with no punctuation.
Author Response

(The authors gave the same response as above.)

Round 2
Reviewer 1 Report
Comments and Suggestions for Authors
Thank you for giving me the opportunity to re-evaluate the article. It has improved a lot and is now much clearer and more readable. The authors have resolved all my questions well with the exception of one that remains unanswered:
It is not that the presentation of these data tends to distract from the correct readability of the manuscript. It is that you have indicated that you have used the PRISMA statement to carry out your review, and you have not considered several of the points included in the checklist. This is not a question of correct or incorrect readability. The points that you have not developed and which are the ones I was asking for are:
In the methodology section (2.6 Study Quality Assessment):
Point 11, Evaluation of the risk of bias of the studies. You should specify the methods used to assess the risk of bias in the included studies, including details of the tool(s) used, how many reviewers assessed each study and whether they worked independently, and if applicable, details of the automation tools used in the process.
Item 14, Assessment of reporting bias. Describe any methods used to assess the risk of bias due to missing results in a synthesis (arising from reporting bias).
Item 15. Certainty assessment. Describe any methods used to assess the certainty (or confidence) in the body of evidence for an outcome.
In the section on results:
Item 18, Risk of Bias in Studies. Present the risk of bias assessments for each included study.
Item 21, Reporting bias. Present assessments of the risk of bias due to missing results (arising from reporting biases) for each synthesis evaluated.
Item 22, Certainty of Evidence, Present assessments of certainty (or confidence) in the body of evidence for each outcome assessed.
Comments on the Quality of English LanguageMinor editing of English language required
Author Response
I am deeply grateful to the reviewer, who has allowed us to improve our work. The response to his comments is attached in the Word document.
